# The Influence of Phytosociological Cultivation and Fertilization on Polyphenolic Content of *Menthae* and *Melissae folium* and Evaluation of Antioxidant Properties through In Vitro and In Silico Methods

**DOI:** 10.3390/plants11182398

**Published:** 2022-09-14

**Authors:** Emanuela Alice Luță, Andrei Biță, Alina Moroșan, Dan Eduard Mihaiescu, Manuela Ghica, Dragoș Paul Mihai, Octavian Tudorel Olaru, Teodora Deculescu-Ioniță, Ligia Elena Duțu, Maria Lidia Popescu, Liliana Costea, George Mihai Nitulescu, Dumitru Lupuliasa, Rica Boscencu, Cerasela Elena Gîrd

**Affiliations:** 1Faculty of Pharmacy, University of Medicine and Pharmacy “Carol Davila”, Traian Vuia 6, 020956 Bucharest, Romania; 2Department of Pharmacognosy & Phytotherapy, Faculty of Pharmacy, University of Medicine and Pharmacy of Craiova, Petru Rareș 2, 200349 Craiova, Romania; 3Department of Organic Chemistry “Costin Nenițescu”, Faculty of Chemical Engineering and Biotechnologies, University of Politehnica, Gheorghe Polizu 1-7, 011061 Bucharest, Romania

**Keywords:** *Mentha piperita* L. leaves extract, *Melissa officinalis* L. leaves extract, phytosociology, polyphenolic content, FT-ICR MS, UHPLC-MS, antioxidant activity, molecular docking, sirtuins

## Abstract

Since medicinal plants are widely used in treating various diseases, phytoconstituents enrichment strategies are of high interest for plant growers. First of all, we investigated the impact of phytosociological cultivation on polyphenolic content (total flavonoids—TFL, and total polyphenols—TPC) of peppermint (*Mentha piperita* L.) and lemon balm (*Melissa officinalis* L.) leaves, using spectrophotometric methods. Secondly, the influence of chemical (NPK) and organic (BIO) fertilization on polyphenolic content and plant material quality was also assessed. Dry extracts were obtained from harvested leaves using hydroethanolic extraction solvents for further qualitative and quantitative assessment of phytoconstituents by FT-ICR MS and UHPLC-MS. Furthermore, the antioxidant activity of leaf extracts was determined in vitro using DPPH, ABTS and FRAP methods. Molecular docking simulations were employed to further evaluate the antioxidant potential of obtained extracts, predicting the interactions of identified phytochemicals with sirtuins. The concentration of polyphenols was higher in the plant material harvested from the phytosociological culture. Moreover, the use of BIO fertilizer led to the biosynthesis of a higher content of polyphenols. Higher amounts of phytochemicals, such as caffeic acid, were determined in extracts obtained from phytosociological crops. The antioxidant activity was dependent on polyphenols concentration, more potent inhibition values being observed for the extracts obtained from the phytosociological batches. Molecular docking studies and MM/PBSA calculations revealed that the obtained extracts have the potential to directly activate sirtuins 1, 5 and 6 through several polyphenolic compounds, such as rosmarinic acid, thus complementing the free radical scavenging activity with the potential stimulation of endogenous antioxidant defense mechanisms. In conclusion, growing medicinal plants in phytosociological cultures treated with biofertilizers can have a positive impact on plant material quality, concentration in active constituents and biological activity.

## 1. Introduction

The use of medicinal plants in various types of ailments has attracted many research efforts into their cultivation. Enrichment in active principles, generation of a larger mass of quality plant products can become major concerns for medicinal plant growers. In this context, studying combinations of medicinal plants belonging to the same family or to different classes may constitute new and growing research directions. Rotational cultivation or intercropping of medicinal plants with various food species may be another starting point for various research projects aimed at enriching plant products with biologically active compounds. Rotational cropping has already been practiced for a long time and on a large scale in agriculture. Intercropping refers to the simultaneous cultivation of two or more species in the same area or plot during a growing season. In an intercropping study of mint and *Vicia faba* L., it was found that a higher amount of volatile oil was produced in mint, and that the dominant compounds, menthol and menthone, did not positively influence the amount of generated biomass [1]. In another intercrop of mint and soybean, a positive influence on the quality of mint volatile oil was observed, with menthol being produced in higher amounts [2]. The main purpose of these types of experimental crops is to limit the aggressiveness of external factors, such as plants (weeds) or animal (insects etc.) pests, applying possible quantifiable treatments, in order to obtain a higher plant mass production compared to monoculture systems. However, when fennel and dill were intercropped, a large amount of biomass was provided only by *Anethum graveolens* L., also acting as the dominant species [3].

Based on all these aspects, the paper presents the initiation of a phytosociology study, in which the aim was to cultivate two medicinal species with extensive use in phytotherapy, *Mentha piperita* L. (mint) and *Melissa officinalis* L. (melissa, lemon balm), in common (phytosociological) crops. One of the scopes of the present study started from a simple premise: the possibility that there might be either positive or negative differences in the biosynthesis of certain active chemical constituents or the supply of plant mass for different types of medicinal species cultivated in common cultures

*Mentha piperita* L. (mint) is a well-known species in eastern and northern Europe, cultivated continuously at a worldwide scale. Mint leaves (*Menthae folium*) are widely used in phytotherapy for their digestive tonic (volatile oil and bitter principles), choleretic-cholagogue and antispasmodic (flavones, polymethoxylated flavones, volatile oil esters, caffeic acid, chlorogenic acid), antidiarrheal (tannin and volatile oil), anti-infectious (tannin, volatile oil), antiemetic (menthol in the volatile oil, which causes slight anesthesia of the gastric mucosa), antipruritic (menthol in the volatile oil), mildly sedative (esters in the volatile oil), antifungal, antiviral (active on herpes virus, due to volatile oil, rosmarinic acid), analgesic (volatile oil) and antioxidant (polyphenols) activities [4,5,6,7,8,9,10].

*Melissa officinalis* L. (melissa, lemon balm), a species of the *Lamiaceae* family such as mint, is associated in phytotherapy for its sedative, antispasmodic (through aldehydes and esters in the volatile oil), choleretic-cholagogue (through caffeic acid, chlorogenic acid, sea principles), antiherpetic, antimicrobial (through rosmarinic acid and aldehydes in the volatile oil), immunomodulating (through polyphenolic derivatives and volatile oil) and antioxidative (through polyphenols) activities [11,12,13,14].

The main scope of the present study was to assess whether it is possible to generate compatible batches of medicinal plants that grow together and produce higher amounts of polyphenolic derivatives and plant mass. More specifically, the objectives of the study were as follows: to monitor mint and melissa crops from common batches and to compare with single-component batches in terms of variation in polyphenol content (flavones and total polyphenols); to compare the polyphenol content in plant products from batches supplemented with biofertilizers and chemical fertilizers; to obtain dry extracts in which the aim was to quantify by spectrophotometric and HPLC methods the polyphenolic derivatives; and to evaluate their antioxidant action using both in vitro and in silico methods. The in vitro methodology consisted in the assessment of the direct antioxidant activity using free radical scavenging assays. The in silico studies were used to evaluate the potential of identified phytochemicals to stimulate the endogenous antioxidant defenses through indirect mechanisms of action, such as stimulating the enzymatic activities of sirtuins.

Sirtuins are a class of nicotinamide-adenine dinucleotide (NAD)-dependent deacetylases which were shown to play important roles in oxidative stress, inflammation and ageing. There are seven known sirtuin isoforms (SIRT1-7) that are localized either in the cytoplasm (SIRT2), mitochondria (SIRT3, SIRT4, SIRT5) or nucleus (SIRT1, SIRT6, SIRT7) [15,16]. Pharmacological activation of sirtuins can have beneficial effects in oxidative stress-related disorders, and several phytochemical compounds were found as up-regulators, such as curcumin, resveratrol, fisetin and quercetin [17,18,19,20]. Therefore, in this present study we also aimed to investigate the potential stimulatory activity of the detected polyphenols on three sirtuin isoforms (SIRT1, SIRT5 and SIRT6) by using molecular docking simulations.

## 2. Result

### 2.1. Quantitative Chemical Analysis of Plant Material from Single and Phytosociological Crops

The results obtained from the quantitative chemical determinations are presented in Table 1.

As expected, from a quantitative, chemical point of view, the plant products from the four batches had variable contents in secondary metabolites. It was found that the amount of polyphenolic derivatives in the phytosociological crops is significantly higher than in the control crops. For mint, the concentration of TFL is twice as high in the phytosociological batch (25.87 ± 5.766 mg/g) compared to the control batch (10.11 ± 2.526 mg/g) and TPC are 1.10 times higher in the phytosociological group (101.43 ± 19.329 mg/g). Lemon balm plant material coming from the phytosociological crop contained 1.4 times more flavones and 1.6 times more TPC when compared to the control batch. Statistically, as can be seen in detail in relation to the level of significance in the boxplot graphs (Appendix A), it can be observed that simple main effects analysis indicated that the common crop is statistically different from control crop (*p* = 0.0002). Post hoc analysis indicated that there was a visible difference in the case of TPC only for lemon balm.

### 2.2. The Influence of Fertilization on the Quality of Soil and Plant Raw Materials

Productive agricultural soils were used for the growth of the studied cultures. Unfortunately, we did not possess previous data on soil quality for comparison with our results. In the first stage we performed a soil analysis in order to choose the optimal type of fertilizer. After applying both chemical and the biological fertilizers, we established two new batches of mint and lemon balm on the two types of fertilized soils. The last stage consisted in collecting the leaves from both species and assessing the influence of fertilization on micro- and macroelements content. The composition of the two fertilized soils was also analyzed. All the obtained data are presented in Table 2.

In general, most cultivated plants prefer neutral or weakly alkaline soils (pH = 6.3–7.5). The NPK-fertilized soil has a slightly lower alkalinity, its pH decreasing from 8.06 in the initial crop to 7.38, possibly due to slightly acidic constituents. In the crop treated with organic fertilizer, the alkalinity is almost the same as in the control crop. However, a slight increase in humus concentration was observed for both batches, which can be explained by the compositions of the two types of fertilizers. Nonetheless, there are no significant differences between the fertilized batches regarding this aspect.

We found that microelements had variable concentrations in the analyzed samples, the use of fertilizers leading to the increase in trace elements. The total nitrogen concentration is higher in the M NPK crop compared to the mint organic crop. Total phosphorus was in approximately equal amounts in M Bio and ML Bio and was higher in the M NPK. Potassium concentration decreased in the fertilized batches, and the highest amounts were found in the ML F batch. The concentration in calcium in the M NPK group was almost identical with the values measured in the M F group; however, it was found to be lower in the M Bio group. Moreover, zinc content increased in the NPK-fertilized crop by 2-fold, and in the organic fertilized crop by 1.6-fold. An interesting decrease in iron concentration was noticed for the organic fertilized crop (0.7-fold lower than in the control batch). In the case of manganese, its concentration increased by 1.5-fold in the NPK-fertilized crop. All these fluctuations in soil quality and trace element concentrations are due to the different chemical composition of the two fertilizers.

An increase in the majority of values for assessed parameters can be easily observed for the plots coming from NPK-fertilized soils. It should be noted that plant species grown in soil with NPK fertilizer are also more developed (Figure 1).

### 2.3. The Influence of Fertilizers on the Biosynthesis of Polyphenols

The results of quantitative chemical determinations on plant products harvested from fertilized soils are shown in Table 3 and were compared with control crops (from unfertilized soil).

Some differences were observed in the content of active principles depending on the type of used fertilizer. In the case of the chemical fertilizer (NPK), there was a slight increase in the concentration of the three types of active principles determined in mint and lemon balm. Moreover, an interesting observation was the significant increase in concentrations in the batch grown with organic fertilizer. In mint crop, the concentration in flavones increases almost 1.3-fold compared to the NPK batch, and for lemon balm, the concentration of flavones almost doubled, while there was a notable increase for the other active principles. Compared to the plots grown without fertilizers, the use of both chemical and organic fertilization lead to a higher production of polyphenolic compounds. Although the medicinal species from NPK-fertilized soil were more developed, the concentrations of active ingredients were actually lower. Statistical analysis showed that there were differences between sample crops only for lemon balm (Appendix A). Post hoc tests revealed only for lemon balm that the bio crop is significantly different from the common crop (*p* = 0.031) and the NPK crop (*p* = 0.014), but there were no significant differences between the common and NPK crops (*p* > 0.05).

### 2.4. Phytochemical Analyses of Dry Plant Extracts

The results obtained from spectrophotometric determinations on plant extracts obtained from the phytosociological and control crops are presented in Table 4. It should be noted that the dried extracts were powdery and homogeneous, while the color and odor were characteristic to the plant products from which they were obtained.

The analyzed results revealed that the dry extracts were enriched in polyphenolic compounds and concentrations varied within wide limits. Interestingly, phytosociological batches showed higher contents than those observed for controls. In mint, for instance, the concentration in flavones was 1.5-fold higher in the extract obtained from the plant product grown in the common batch and the total polyphenols were 1.2-fold higher in comparison with the control batches. In lemon balm, the total polyphenol concentration is 1.7-fold higher in the phytosociological crop.

### 2.5. FT-ICR MS (Fourier-Transform Ion–Cyclotron-Resonance High-Resolution Mass Spectrometer)

Data obtained from FT-ICR MS, ESI+ and ESI− analyses are presented in Table 5. The recorded spectra for ESI+, ESI− and the rest of the obtained spectra can be found in Appendix A. It can be noted that, depending on the type of ionization, there was a slight shift in atomic masses.

By this method, polyphenolic compounds were identified in plant extracts. In consequence, negative ionization allowed the identification of a wider spectrum of polyphenolic compounds in all types of analyzed crop extracts. Through positive ionization, protocatechuic acid could not be detected in any type of extract, while rutin and *p*-coumaric acid were detected only in extracts obtained from lemon balm.

Mass spectra for polyphenolic compounds such as caffeic acid from mint and lemon balm samples are presented in the Appendix A.

### 2.6. UHPLC-MS (Ultra-High Performance Liquid Chromatography-MS)

Results after performing chromatographic analysis of the polyphenolic derivatives from the four extract types are presented in Table 6, and the representative chromatograms for each type of extract are shown in Figure 2.

The quantitatively determined polyphenolic compounds varied within broad limits depending on the type of extract used in the analysis. Although ferulic acid was identified by FT–ICR, the phytochemical could not be further quantified by this method, its concentration being possibly below the detection limit. Higher concentrations of protocatechuic acid were found in the lemon balm extracts compared to mint extracts, caffeic acid concentration was 15 times higher in the melissa extract obtained from the phytosociological crop in comparison with the control; the concentration of quercetin in the mint extract obtained from the phytosociological batch was almost 20 times higher in comparison with the control crop; kaempferol was not quantified in the mint extract obtained from the control crop, and in all other extracts the concentration was low; the concentration of rosmarinic acid found in lemon balm phytosociological crop was 1.26 times higher than the values recorded for mint phytosociological crop.

### 2.7. Evaluation of Antioxidant Activity

#### 2.7.1. In Vitro Antioxidant Assays

The antioxidant effects observed for the tested extracts were directly correlated with the concentration of secondary metabolites (Table 4). IC50 of Vitamin C was determined by DPPH method and its value was 0.0165 mg/mL (Appendix A), IC50 of trolox was determined by ABTS method and its value was 0.0330 mg/mL (Appendix A) and IC50 of FeSO_4_ was determined by FRAP method and its value was 0.1028 mg/mL (Appendix A). Data comparison revealed that the substances that generated the strongest antioxidant activities were found in MLF E (lowest IC50 value by all three methods, compared to the other extracts). It is especially noteworthy that the IC50 values for MLF E were, among the assessed extracts, substantially closer to the antioxidant values of the used control, which emphasizes MLF E’s superior antioxidant action over the other samples. All the analyzed extracts contained significant amounts of total polyphenols, with high concentrations for MF E and MLF E, and moderate concentrations for MM E and MLM E. High amounts of phenolic acids were also found in their composition, high concentrations being observed for MF E and MLF E, and moderate concentrations for the other extracts (Table 7).

Furthermore, it is crucial to assess the relationships between the antioxidant action of the obtained extracts with TFL and total polyphenol content. The values of the Pearson coefficient (r) are negative in all cases, which explains the inverse correlation between the data (the higher the amount of active principles, the lower the IC50 value of the extracts and therefore the stronger the antioxidant action). A moderate correlation is observed between the TFL content and the IC50 values determined by the ABTS and DPPH methods (|r| is between 0.40 and 0.69), but a very strong correlation was recorded for the results obtained by the FRAP method (|r| > 0.900)—Appendix A.

The content of total polyphenols (TPC) in direct correlation with the antioxidant activity of plant extracts, the compared data (TPC concentration vs. IC50) showing a moderate correlation in the case of the DPPH and ABTS methods and a strong correlation for the FRAP method—Appendix A.

It was observed that the IC50 values determined by the FRAP method proved to be much better correlated with the content of active principles (TFL and TPC) than those provided by other assaying methods. The evaluation of the antioxidant action by the three methods (DPPH, ABTS, FRAP) is consistent with the results obtained for the determination of polyphenol content. The high concentrations of polyphenols in the extracts obtained from the plant products of the phytosociological crops could also explain their significantly higher antioxidant action.

#### 2.7.2. In Silico Studies

A set of 11 polyphenolic ligands were docked into the binding sites of three sirtuin isoforms to evaluate the potential of such compounds to act as direct activators. The implemented docking protocol was successfully validated, and the predicted binding poses superposed on the experimental conformations are shown in Appendix A. Only slight variations in ligand orientation were observed for all four positive controls. The SIRT1 and SIRT5 activator resveratrol showed a binding energy of −9.332 kcal/mol and 0.549 ligand efficiency for SIRT1, and a much higher binding energy of −5.220 for SIRT5 (0.307 ligand efficiency). The SIRT6 activator quercetin had a binding energy of −6.899 kcal/mol and 0.314 ligand efficiency, while the SIRT6 inhibitor catechin gallate, a quercetin derivative, had a docking score of −8.732 kcal/mol and 0.273 ligand efficiency. Although catechin gallate is structurally similar to quercetin and shares the same binding pocket, previous studies showed that the differences in ligand and protein sidechain orientations within the binding site are responsible for a total shift in biological activity [21]. Thus, the studies polyphenols were docked into both protein conformations, in order to discriminate between potential stimulatory and inhibitory activities.

The binding energies and ligand efficiencies obtained after the docking simulations are shown in Table 8, while the predicted dissociation constants are presented in Appendix A. The binding energies after docking on SIRT1 ranged from −9.827 kcal/mol to −5.858 kcal/mol, with a mean value of −8.430 ± 1.337 kcal/mol. For SIRT5, docking scores varied between −8.764 and −5.720 kcal/mol, with a mean of −7.217 ± 1.104 kcal/mol. Binding energies after docking on the activator-specific conformation of SIRT5 ranged from −7.775 to −5.721 kcal/mol (−6.698 ± 0.602 kcal/mol), while energies for the inhibitor-specific receptor conformation were between −8.287 and −5.703 kcal/mol (−6.774 ± 0.860 kcal/mol).

When compared to the positive controls, we found that four compounds exhibited higher binding affinities for SIRT1 (rutin, rosmarinic acid, luteolin and kaempferol), all polyphenols had higher affinities for SIRT5 (although only eight had better ligand efficiencies), three ligands showed better energies for SIRT6 as potential activators (rutin, rosmarinic acid, *p*-coumaric acid) and no compounds had better affinities as potential inhibitors, but nine compounds showed better ligand efficiencies. Interestingly, quercetin showed a slightly better binding energy for the SIRT6 inhibitor binding site, although the ligand efficiencies were practically equal. Moreover, luteolin had a better binding affinity for the same binding site, although previous data indicate that luteolin is a SIRT6 activator, rather than an inhibitor [21]. Thus, the docked conformation should be a better indicator of a potential stimulatory or inhibitory activity, rather than the binding energy. On the other hand, isoquercitrin had a much better binding affinity for the SIRT6 activator pocket and was proven to stimulate SIRT6 activity [21].

Regarding the molecular interactions between docked ligands and target proteins, we chose to discuss the predicted interactions for one particular compound, rosmarinic acid, which showed both good docking scores and ligand conformations. Moreover, rosmarinic acid had predicted dissociation constant values of nanomolar range for SIRT1 (86 nM0, while the potencies for other isoforms were of 0.749 µM for SIRT5, 1.999 µM for SIRT6 as activator and 0.842 µM for SIRT6 as inhibitor. Rosmarinic acid formed four hydrogen bonds with SIRT1 binding site, involving residues Asn226, Glu230 and Phe414. Moreover, the ligand formed one additional hydrogen bond with Lys3, which is part of the peptide substrate, thus stabilizing the sirtuin-substrate complex. Furthermore, rosmarinic acid formed hydrophobic pi–alkyl interactions with Pro212, Leu215, Arg446 and Pro447 and van der Waals interactions with other residues (Figure 3a,b).

The complex between rosmarinic acid and SIRT5 is stabilized by hydrogen bonds with Gln140, Asp143 and His158. Two arginine residues are involved in one salt bridge with the carboxylic moiety and one pi–cation interaction with the phenyl ring. Hydrophobic interactions such as pi–alkyl interactions and weak van der Waals forces are observed with other residues within the binding pocket. Unfortunately, one unfavorable acceptor–acceptor interaction was formed between one hydroxyl group and Asp143 (Figure 3c,d).

Rosmarinic acid showed a better binding energy for the binding pocket specific to the SIRT6 inhibitor catechin gallate. However, rosmarinic acid interacted fairly well with the pocket specific to the SIRT6 activator quercetin. Regarding the quercetin binding pocket, rosmarinic acid formed several polar interactions such as three hydrogen bonds with Thr84 and Tyr257, and one hydrogen bond with a water molecule. A pi–pi stacked interaction was formed with Phe82 and several other van der Waals interactions with other residues and a water molecule (Figure 4a,b). The docking experiment revealed that rosmarinic acid occupied a binding subpocket proximal to the inhibitor binding site. The interaction between rosmarinic acid and the inhibitor-specific crystal structure of SIRT6 was characterized by two hydrogen bonds with Arg65 and ADP-ribose (AR6401). Noteworthy is the fact that the interaction with ADP–ribose was not reported for the inhibitor catechin gallate, which indicated that the predicted conformation of rosmarinic acid might not correspond to a potential SIRT6 inhibitory activity. Moreover, the ligand forms an attractive charge interaction with Lys160, a pi–pi stacked interaction with Trp188 and several van der Waals interactions. On the other hand, the phytochemical three types of unfavorable interactions: unfavorable negative charge interactions (Asp187), and unfavorable donor–donor (ADP–ribose) and acceptor–acceptor interactions (Pro10, Figure 4c,d).

The free energies of binding obtained after performing MM/PBSA (molecular mechanics Poisson–Boltzmann surface area) calculations on the last snapshot of the 1 ns molecular dynamics simulations are shown in Appendix A. The performed analysis revealed that rosmarinic acid exhibited much lower free binding energies than two out of three positive controls. Thus, the predicted ligand showed higher binding affinities after 1 ns of simulation than resveratrol for both SIRT1 and SIRT5 isoforms, and higher than SIRT6 activator quercetin. On the other hand, rosmarinic acid had a higher binding energy than the SIRT6 inhibitor catechin gallate. The lowest energy was recorded for the interaction with SIRT6, followed by SIRT5 and SIRT1, respectively. Therefore, the free energy of binding calculations further strengthened the hypothesis that rosmarinic acid may have the potential to activate SIRT6, rather than inhibiting the isoform, while also possibly acting as SIRT1 and SIRT5 direct activators.

## 3. Discussion

In this current study, *Menthae folium* and *Melissae folium* plant products harvested from species grown in common (phytosociological) crops were analyzed in comparison with control crops. The crops were grown on an agricultural field, in Teleorman county, near Turnu Măgurele, Romania. The obtained results are in agreement with previously published data from our research [22,23], when the raw material for 2019 and 2020 was analyzed. Growing the two species in common batches is beneficial not only for their horizontal and vertical development or generation of a large mass of plant material, but also for the biosynthesis of a significantly higher amount of polyphenols.

The quantitative chemical profile of the plant raw materials was determined by spectrophotometric methods and polyphenolic derivatives were assessed (flavones and total polyphenols). Although spectrophotometric methods cannot be considered selective methods of analysis (possible interference with other types of constituents), they provide information regarding the polyphenol content, and they are frequently used and described in European Pharmacopoeia 10th edition (Chapter 8.8.14. Tannis in herbal drugs; dosage of flavones in various plant product monographs, e.g., *Betulae folium*—expressed in hyperoside; *Sambuci flos*—expressed in isoquercitroside).

From a statistical point of view, it was found that there was an interdependence between the content of active principles and the batch from which the plant raw material came from (single crop vs. phytosociological crop). Statistical differences were observed in the content of total polyphenols only for lemon balm.

In order to determine the influence of fertilizers on polyphenol biosynthesis, assessments were also performed on plant products harvested from species grown on the farmland where one chemical (NPK) and one organic fertilizer were used.

We consider weak alkaline soil to be beneficial to the culture, given the quantitative chemical results presented above. Soil humus is a complex mixture of compounds resulting from the transformation of organic and microbial residues. With a concentration of almost 3% in humus, we can consider it an average soil enriched in these natural complexes. The three macronutrients in the soil, nitrogen, phosphorus and potassium are very important for plant development. The presence of nitrogen and phosphorus in the soil is important for stimulating the root growth of medicinal plants, and for nutrient uptake. Potassium increases plant mass production and improves their quality [24]. Soil trace element content is correlated with soil quality. The crops were grown in an ecological area of Teleorman county, for this reason we consider that the soil has low concentrations of the analyzed trace elements.

The comparison between soil fertilization with organic and NPK fertilizers was performed to assess their influences on the amount of active ingredients produced by the respective plant raw materials. Although the crop species from the soil fertilized with NPK were better developed and generated a greater amount of plant raw material, the polyphenols were biosynthesized in much lower concentrations. For example, M Bio generated 1.3-fold more TFL compared to M NPK and 1.5-fold more TPC. In the lemon balm crops, the highest variation in active compounds was observed for TFL. In the BIO fertilized crop, the concentration was 1.7-fold higher compared to the ML NPK crop, while the TPC was 1.18-fold higher. When compared to the unfertilized batches, the most important differences were observed for the crops where the BIO fertilizer was used.

The use of fertilizers for mint samples was also reported by other researchers: Hend S et al. [25] investigated the influence of fertilizer types on volatile oil production. Sheykholeslami Z. et al. [26] found that soil treatment with different types of fertilizers was beneficial to the development of *Mentha piperita* L. species, a higher quantity of plant product was generated, and vertical growth was also significantly higher [27]. According to studies by Marin N. et al. [28], on a field located in Teleorman county, a successive fertilization of the soil did not lead to an overload with trace elements, a finding also observed during our research on soil enriched with the two types of fertilizers. For instance, the concentration of manganese quantified in NPK-fertilized soil is much lower compared to the data found in the literature [24].

Dry extracts were also obtained from samples retrieved from species grown in fertilizer-free plots. For these extracts, we determined the polyphenolic profile by spectrophotometric, FT–ICR MS and UHPLC–MS methods, and we also investigated the antioxidant activity.

Growing in phytosociological crops can be a practice that can be extended to medicinal plants. Enhancement of horizontal and vertical development, and generation of a larger quantity of plant mass enriched in active ingredients can be the basis for further studies, and the relevant findings could be transferred to indigenous producers of medicinal plant crops. Plant products from common (phytosociological) batches have a higher amount of polyphenols, which varies greatly depending on the nature of the plant raw material. Extractions of polyphenols from plant products were made in 50% (for mint) and 70% (for lemon balm) ethanol, since previous studies reported that these concentrations were shown to yield the best results [22,23]. At the same time, we aimed to use solvents that are more environmentally friendly and do not generate toxic metabolites.

Mint and lemon balm are species belonging to the same family (*Lamiaceae*), are aromatic plants, and can be positively influenced (as shown for polyphenol content) by being cultivated in common crops. Hydroethanolic plant extracts were prepared from common and control batches.

Extracts obtained from plant products harvested from the common crops had a significantly higher polyphenol content compared to the control crops. There was a high accumulation in total polyphenols compared to flavones; e.g., in the mint extract obtained from the common batch products, the concentration in total polyphenols was 4.7-fold higher compared to flavones, and in lemon balm there were 7.3-fold more total polyphenols compared to flavones.

FT–ICR MS and UHPLC–MS analysis allowed the identification and quantification of polyphenol content; increased concentrations of polyphenols were found in lemon balm for caffeic acid, chlorogenic acid and luteolin. The *Melissa* extract obtained from the plant product harvested from the common (phytosociological) crop contained 3.7 times more caffeic acid compared to mint harvested from the same crop, 4.5 times more chlorogenic acid and 6.3 times more luteolin. In the control batches, the differences between the two species in the active principles content was much smaller. Based on the obtained results, we can conclude that the association of the two species in phytosociological culture leads to an enrichment in polyphenol-type phytoconstituents.

Although it was found in small quantities in the analyzed batches, protocatechuic acid (3,4-dihydroxybenzoic acid) is of high importance, since this phytochemical is considered to be a perfect peroxyl radical scavenger in the polar medium of aqueous solutions, and a relatively good free radical scavenger in the non-polar medium of lipid solutions. It is able to attenuate oxidative stress by increasing glutathione peroxidase (GSH-Px) and superoxide dismutase (SOD) activity, as well as reducing xanthine oxidase (XOD) and NADPH oxidase (NOX) activity and malondialdehyde (MDA) concentrations [29]. Phytotherapy supplementation with extracts rich in rutin (quercetol 3-rhamnoglucoside) is beneficial, given the multiple therapeutic virtues it presents [30], such as preventing the oxidation of LDL-cholesterol involved in atherosclerosis [31]. Furthermore, rutin has been shown to be effective in terms of free radical scavenging capacity (presence of the four phenolic hydroxyl groups in the chemical structure), may be a potential hydrogen donor, and has been shown to have a higher DPPH radical scavenging capacity than vitamin C [32,33,34]. Luteolin, a flavone derivative found in a wide variety of vegetables and fruits, with an average daily intake of 0.01–0.20 mg/day [35], is implicated in a variety of therapeutic effects at the cellular level (cardioprotective, hypocholesterolemic, antitumor, anti-inflammatory) due to its antioxidant effects [36,37,38,39]. Caffeic acid (3,4-dihydroxycinnamic acid) has been shown to be a protector of alpha-tocopherol in low-density lipoprotein (LDL) [40], and is a compound with a clearly superior antioxidant activity against LDL-cholesterol oxidation, when compared to *p*-coumaric and ferulic acid [41,42]. Rosmarinic acid, a phenolic compound derived from hydroxycinnamic acid, is frequently found in species of the *Lamiaceae* family, and is recognized for its antioxidant, anti-inflammatory, hepatoprotective, cardioprotective and neuroprotective activities [43].

The intensity of the antioxidant action is dependent on the polyphenol content for each type of the analyzed extracts. DPPH radical inhibition is 0.7-fold higher for common batch lemon balm extract compared to mint extract. Furthermore, a 0.8-fold higher reduction of the non-biological radical ABTS and 0.9-fold higher antioxidant power for ferric ion reduction were observed. Considering that, from a mathematical point of view, the Pearson correlation coefficient has certain intervals that express the degree of correlation between the experimental data sets, where we obtained values lower than 0.900 for the r coefficient, we can quantify the existing relationship between the analyzed data even if statistically we cannot extrapolate what we observed to the entire target population.

The antioxidant potential of the studied extracts was also evaluated using in silico methods. Molecular docking simulations were carried out to investigate the potential interactions between identified polyphenols and three sirtuin isoforms (SIRT1, SIRT5 and SIRT6), as a means to predict the stimulatory activity on endogenous antioxidant defense mechanisms. Moreover, the available crystal structure of SRT6 in complex with a natural inhibitor (catechin gallate) allowed us to discriminate between activators and inhibitors. Interestingly, none of the docked compounds showed conformations in the catechin gallate allosteric binding site that resembled the orientation of the positive control, even though caffeic acid, chlorogenic acid, ferulic acid, kaempferol, luteolin, quercetin, rosmarinic acid and rutin had better binding affinities for this receptor structure. Except for protocatechuic acid, all docked ligands showed rather high binding affinities for SIRT1 and could act as direct SIRT1 activators by stabilizing the complex between the enzyme and substrate. Furthermore, quercetin, ferulic acid, caffeic acid ethanolamide, chlorogenic acid, kaempferol, luteolin and protocatechuic acid were shown to up-regulate SIRT1 activity in various experimental settings [44,45,46,47,48]. Moreover, other authors hinted that rosmarinic acid, which we discussed in more detail, demonstrated anti-inflammatory and anti-apoptotic effects in a mouse model of nonalcoholic steatohepatitis, possibly due to stimulating SIRT1-mediated pathways [49]. On the other hand, quercetin can directly inhibit SIRT1 activity [21]. Rosmarinic acid, rutin and isoquercitrin showed particularly high binding affinities for SIRT5. No data were found in the literature regarding the first two compounds, while isoquercitrin was shown to be active only on SIRT6. The same study highlighted that SIRT6 is activated also by luteolin and quercetin [21].

Predicted dissociation constants were also calculated after molecular docking simulations. The docking experiments revealed that isoquercetin, kaempferol, luteolin, quercetin, rosmarinic acid and rutin had Kd values lower than 1 µM for SIRT1, while only isoquercetin, rosmarinic acid and rutin had Kd values within the same range for SIRT5. However, as potential SIRT6 activators, only *p*-coumaric acid, rosmarinic acid and rutin had Kd values ranging between 1 and 10 µM, the lowest being 1.999 µM for rosmarinic acid. Among these compounds, rutin, luteolin, rosmarinic acid, quercetin and isoquercetin were found in relatively high concentrations in both phytosociological plant extracts. Another phytochemical detected in high amounts was caffeic acid, for which docking simulations showed relatively lower potency values, but remarkably high ligand efficiencies. Furthermore, the aforementioned polyphenols could act as potent antioxidants through both direct and indirect mechanisms, in a synergistic manner: by acting as free radical scavengers and promoters of antioxidant defenses through direct stimulation of sirtuins.

A more detailed analysis of the molecular interactions between rosmarinic acid and sirtuins supported the potential to directly activate SIRT1, 5 and 6, rather than inhibit SIRT6. These observations were also strengthened by the MM/PBSA binding free energy calculations, which revealed that rosmarinic acid had markedly lower binding energies than SIRT1 and SIRT5 activator resveratrol and SIRT6 activator quercetin, while exhibiting a higher energy than SIRT6 inhibitor catechin gallate. Among the three isoforms, rosmarinic acid showed the lowest binding free energy for SIRT6. Therefore, the radical scavenging activity of the studied extracts might be complemented in vivo by the up-regulation of sirtuins activity by rosmarinic acid and other phytochemical constituents, thus leading to an enhanced antioxidant protection in various diseases.

## 4. Materials and Methods

### 4.1. Establishing the Quality of Plant Raw Materials

We determined the quality of the raw materials using classic and common spectrophotometric methods, cited in the literature and frequently applied in the assessment of these types of active principles. Since plant extracts are mixtures of many complex substances, these determinations are frequently used. Even the UHPLC MS method confirmed the identity of the globally assessed compounds in the plant raw materials.

#### 4.1.1. Plant Materials, Reagents and Equipment

Back in 2018, we started to cultivate in common (phytosociological) crops two medicinal plants, *Mentha piperita* L and *Melissa officinalis* L. Peppermint and lemon balm were planted using experimental plots with following characteristics: area of 50 cm × 300 cm, 400 cm between batches, 30 cm between seedlings and 5 seedlings per group. The cultures were grown in Turnu Magurele City’s suburbs in Teleorman County (43°44′44.16″ Northern latitude, 24°52′53.40″ Eastern longitude), Romania [22]. This area presents average yearly temperatures of 11.5 °C, average monthly temperatures of 23 °C for the warm ones and average monthly temperatures less than 2 °C for the cold ones. It is distinguished by a high caloric potential, high air temperature amplitudes, little precipitation, frequent torrential regime in the summer and frequent drought intervals. In order to avoid being influenced, the control crops were planted apart from the common culture. By comparing each crop with the control batch, the morphological and phytochemical characteristics of the harvest were investigated [23]. In July of each year, the plants were collected and dried in laboratory conditions at the department of Pharmacognosy, Phytochemistry and Phytotherapy of the Faculty of Pharmacy. The study took place during 2018–2021 and the data collected in 2021 are presented in this paper.

Based on previous studies [22], the solvents used for the extraction of active principles were 50% ethanol for mint (MM—peppermint leaves from control crop, MF—peppermint leaves from common crop) and 70% ethanol for lemon balm (MLM—lemon balm leaves from control crop, MLF—lemon balm leaves from control crop). The solvent was chosen to ensure the best possible extraction of phenolic constituents from all the examined leaves. The ethanol used as solvent in this section was purchased from Sigma–Aldrich, Hamburg, Germany.

Approximately 1.000 g of dried leaves from each plant was brought into 50 mL ethanol and then subjected to refluxing for 30 min. On the extractive solution obtained by filtration in a 50 mL flask [22], we performed further analyses presented in the following paragraphs.

#### 4.1.2. Determination of Polyphenolic Content

Total flavonoid content (TFL) and total phenolic content (TPC) were all determined using spectrophotometric techniques.

##### Determination of Total Flavonoids Content (TFL)

The total flavonoids content assay utilized a colorimetric technique based on the reaction between flavonoids and AlCl_3_. From our obtained extractive solutions, we made dilutions of 10:25 mL and, thereafter, 5 different volumes were brought into 10 mL volumetric flasks. Then, 2 mL sodium acetate 100 g/L (Sigma–Aldrich, Hamburg, Germany) and 1 mL aluminium chloride solution 25 g/L (Sigma–Aldrich, Hamburg, Germany) were added. Further, all the volumes were adjusted to 10 mL by adding the same solvent as above. In parallel with the samples to be analyzed, the appropriate control samples were prepared in the same conditions but without sodium acetate and aluminum chloride. After 45 min, the absorbance was measured at 427 nm (Jasco V–530 spectrophotometer, Hachioji, Japan). Rutin (Sigma–Aldrich, Hamburg, Germany) was used as a standard for the linear calibration curve in the concentration range of 5–35 μg/mL with R^2^ = 0.9992. The total flavonoids content (TF) of the extract was expressed as mg rutin equivalents per gram of sample (mg/g) [50].

##### Determination of Total Phenolic Content (TPC)

Total polyphenols (TPC) were determined in accordance with Lamuela–Raventós’s [51] methodology with a few minor adjustments. Same dilution was used and volumes between 0.1 mL and 0.6 mL were brought into 10 mL volumetric flasks and were adjusted to 1 mL by adding distilled water. Then, the volumes were mixed with 1 mL Folin–Ciocalteu’s phenol reagent (Sigma–Aldrich, Hamburg, Germany) and kept at 25 °C for 5–8 min before adding 8 mL sodium carbonate solution 200 g/L (Sigma–Aldrich, Hamburg, Germany). After 40 min in dark conditions, the absorbance was measured at 725 nm (Jasco V–530 spectrophotometer, Hachioji, Japan). The absorbance was measured relative to a blank sample obtained by mixing 1 mL distilled water with 1 mL Folin–Ciocalteu’s reagent and then adjusted to 10 mL by adding sodium carbonate. Tannic acid (Sigma–Aldrich, Hamburg, Germany) was used as a standard for the calibration curve in a linear concentration range of 2–9 µg/mL with R^2^ = 0.999. The total phenolic content (TP) was expressed as mg tannic acid equivalents per gram of sample (mg/g) [50].

### 4.2. The Influence of Fertilizers on the Biosynthesis of Secondary Metabolites

#### 4.2.1. Assessment of Soil Composition and Plant Material

Soil samples have been taken from several areas at a depth of 10–15 cm and the concentrations of micro and macroelements was assessed. These determinations were performed at Physico-Chemical Analysis Laboratory for Soil Sciences, Agrochemistry and Environmental Protection (LAFC) within the National Research-Development Institute for Pedology, Agrochemistry and Environmental Protection (ICPA), represented by Head of Laboratory, Dr. Nicoleta Vrînceanu, as executor of the tests.

The parameters evaluated to determine the quality of control and fertilized soil, also NPK 20–20–20 and Bio–Fertil 20 formulas can be found in Appendix A. Through these methods described there, we determined the presence and concentrations of N, P, K, Ca, Zn, Cu, Fe and Mn.

Moreover, the widely described methods by which the macro- and microelements in the plants’ leaves obtained on these soils were analyzed are presented in the Appendix A.

#### 4.2.2. Determination of the Quality of Extractive Solutions from Fertilized Material

Spectrophotometric methods that were used for the determination of total flavonoid content (TFL) and total phenolic content (TPC) are presented in Section 4.1.2.

### 4.3. Plant Extracts Preparation

Dry leaves from M–ML common and control crops were used for obtaining dry extracts. Based on our previous study [22], the solvent used for the extraction was 50% ethanol for all plants. Exactly 25 g of plant material were used from every crop and were subjected to two consecutive reflux extraction processes: the first extraction used 1.5 L solvent for 30 min, while the second used 750 mL solvent for 30 min. The two extract solutions were mixed and concentrated in a rotary evaporator (Buchi, Vacuum Pump V-700) and then subjected to a lyophilization process (Christ Alpha 1–2/B Braun, BiotechInt, New Delhi, India). The dry extracts were conserved in a glass vacuum desiccator [50]. The samples were marked as follows: MM E (*Mentha* extract from control crop), MF E (*Mentha* extract from common crop), MLM E (*Melissa* extract from control crop) and MLF E (*Melissa* extract from common crop). Each stage of the research includes a presentation of additional tools and experimental setups used in this investigation.

### 4.4. Phytochemical Analysis of Plant Extracts

Spectrophotometric methods used for the quality control of plant extracts are described at 4.1.2. (Determination of Polyphenolic Content). In the case of FT–ICR MS, the technique allows the identification of a minimum of 300 compounds using direct electrospray infusion ionization, without chromatographic separation, depending on the monoisotopic mass in a very short time. The hyphenated method known as Ultra-High Performance Liquid Chromatography (UHPLC–MS) was used to establish the polyphenolic profile of the plant extracts based on non-targeted tandem mass spectrometry (MS–MS). The same method was used for the quantification of selected polyphenolic compounds for each available analytical standard (Sigma–Aldrich, Hamburg, Germany).

#### 4.4.1. Assessment of TFL and TPC

Analytical determination of TFL and TPC was performed according to the method described at 4.1.2. (Determination of Polyphenolic Content).

#### 4.4.2. Identification of Polyphenolic Compounds by FT–ICR MS

FT–ICR MS with 15T superconducting magnet (solar X–XR, QqqFT–ICR HR, Bruker Daltonics) was used for electrospray ionization (ESI) analysis (HR–MS). For negative ESI ionization, the sample was introduced by direct infusion, with a sample flow rate of 120 µL/h, a nebulizing gas pressure (N_2_) of 4 bar at 200 °C, and a flow rate of 7 L/min. The spectra were recorded over a mass range between 46 and 800 uam at a source voltage of 5700 V. For the positive ESI ionization, the sample was introduced by direct infusion, with a sample flow rate of 120 µL/h, a nebulizing gas pressure (N2) of 3.2 bar at 180 °C, and a flow rate of 5 L/min. The spectra were recorded in a mass range between 46 and 800 uam at a source voltage of 5500 V. 

It is well acknowledged that Fourier transform ion cyclotron resonance mass spectrometry (FT–ICR MS) is one of the most effective methods for analyzing organic mixtures at the molecular level. It is frequently possible to identify organic molecules using only the recorded mass-to-charge (*m*/*z*) values due to the ultra-high mass resolution of FT–ICR MS. Organic mixtures such as metabolites [52], vegetal oils [53], wine [54], explosives [55], coal extracts, [56], humic materials [57,58] as well as crude oils [59,60], have all been effectively analyzed using FT–ICR MS. Broadband FT–ICR MS spectra of these types of organic combinations are usually very complicated, with peaks appearing across a large dynamic range [56].

#### 4.4.3. Identification and Quantification of Polyphenolic Compounds by Ultra-High Performance Liquid Chromatography–MS (UHPLC–MS)

Reagents: Protocatechuic (PRO), caffeic (CAF), *p*-coumaric (COU) and ferulic (FER) acids were purchased from Merck (Kenilworth, NJ, USA), chlorogenic (CHL) acid was obtained from Alfa Aesar (Haverhill, MA, USA), rosmarinic (ROS) acid, rutin (RUT) and quercetin (QUE) were acquired from Sigma Aldrich, luteolin (LUT) and kaempferol (KAE) were purchased from Roth (Dautphetal, Germany), while isoquercitrin (ISO) was obtained from HWI Analytik (Rülzheim, Germany). All HPLC gradient grade solvents (water, acetonitrile) were purchased from Merck. The calibration curve for this method with standard chromatogram, and retention times can be found in Appendix A.

### 4.5. Evaluation of Antioxidant Activity

#### 4.5.1. In Vitro Assays

##### DPPH Free Radical Scavenging Activity (2,2-Diphenyl-1-picrylhydrazyl)

According to Celik S.E., the antioxidant activity is influenced by both the properties of the substrate and the polarity of the solvent. The DPPH assay was used to measure the free radical scavenging activity of the plant extracts [61]. Equal amounts of 0.1 g dry extracts were dissolved in 100 mL 50% ethanol for all dry extracts. Ten corresponding volumes of each obtained solution were brought into 10 mL volumetric flasks and were adjusted to 10 mL by adding the same solvent as above. 0.5 mL of each diluted solution was mixed with 3 mL DPPH 0.1 mM radical solution (Sigma–Aldrich, Hamburg, Germany) [62]. The solutions were protected against light for 30 min, and the absorbance was then measured at 515 nm using a spectrophotometer (Jasco, Hachioji, Japan). Ascorbic acid (Sigma–Aldrich, Hamburg, Germany) was used as a reference for the calibration curve in the concentration range of 2–22 µg/mL [50].

The percentage of DPPH^•^ inhibition was calculated using the formula below [63]:(1)$$\%\;\mathrm{DPPH}\;{\mathrm{inhibition}}_{\;}=\frac{A\;\left(blank\right)-A\;\left(sample\right)}{A\;\left(blank\right)}\times 100$$
where: 

*A (blank)* = blank absorbance of DPPH 0.1 mM solution in the absence of extracts (1.00 ± 0.10);

*A (sample)* = sample absorbance of the DPPH solution in the presence of extracts after 30 min.

Based on the established values, inhibition curves (%) were constructed depending on the concentration (mg/mL). Using the linear equations, the IC50 values (mg/mL) were determined for each extract (for the value y = 0.5).

##### ABTS Method of Total Antioxidant Capacity Assessment

Due to the fact that the antioxidant response involves faster reaction kinetics in a pH-independent way, the ABTS assay is regarded as one of the most sensitive assays to evaluate the antioxidant activity of both hydrophilic and lipophilic substances [64,65].

The reaction of ABTS (Sigma–Aldrich, Hamburg, Germany) 7.4 mM solution with potassium persulfate 2.6 mM (K_2_S_2_O_8_—Sigma–Aldrich, Hamburg, Germany) produced the ABTS radical cation (ABTS^•+^), which was then stored at room temperature and in darkness for 16 h before use [66].

Equal amounts of 0.1g dry extracts were dissolved in 100 mL 50% ethanol for every plant extract used in our study. Seven corresponding volumes of each obtained solution were brought into volumetric flasks and adjusted to 10 mL by adding the same solvent as above. 0.5 mL of each diluted solution was mixed with 3 mL ABTS^•+^ solution diluted with ethanol (Sigma–Aldrich, Hamburg, Germany). The solutions were stirred and held in darkness for 6 min [67]. The absorbance was then measured at 734 nm, relative to absolute ethanol, using a spectrophotometer (Jasco, Hachioji, Japan).

The percentage of ABTS^•+^ inhibition was calculated using the following formula:(2)$$\%\;\mathrm{ABTS}\;\mathrm{inhibition}=\frac{A\;(t=0\min)-A\;(t=6\min)}{A\;\left(t=0\;min\right)}\times 100_{,}$$
where:

*A (t =* 0 *min)* = absorbance of the blank sample (ABTS^•+^ solution in the absence of tested samples: 0.70 ± 0.02);

*A (t =* 6 *min)* = absorbance of the vegetal extract (ABTS^•+^ solution in the presence of tested samples).

The concentration of sample needed to scavenge 50% of the ABTS^•+^ free radical, or the IC50 value, was determined by plotting radical scavenging activity against extract concentration (IC—inhibitory concentration). The antioxidant activity of an extract is inversely correlated with the IC50 value.

##### Antioxidant Activity Using FRAP Assay (Ferric Reducing Antioxidant Power Assay)

A modified FRAP assay was used to assess the ferric reducing capacity of plant extracts [65]. The reduction of ferric iron (Fe^3+^) to ferrous iron (Fe^2+^) by antioxidants present in the samples is how the assay determines the antioxidant potential. Blue coloration results from the conversion of ferric iron (Fe^3+^) to ferrous iron (Fe^2+^).

Equal amounts of 0.1 g dry extracts were dissolved in 100 mL 50% ethanol for every plant extract used in our study. Eight corresponding volumes of each obtained solution were brought into volumetric flasks and adjusted to 10 mL by adding the same solvent as above. An amount of 2.5 mL of each diluted solution was mixed with phosphate buffer (pH 6.6, Sigma–Aldrich, Hamburg, Germany) and 2.5 mL K_3_(FeCN)_6_ 1% (Sigma–Aldrich, Hamburg, Germany) before being heated to 50 °C for 20 min. 2.5 mL trichloroacetic acid (Sigma–Aldrich, Hamburg, Germany) was added to each sample. Furthermore, 2.5 mL of distilled water and 0.5 mL FeCl_3_ 0.1% (Sigma–Aldrich, Hamburg, Germany) were added to 2.5 mL of each of the resulting solutions, the samples being left thereafter idle for 10 min. The change in the absorbance at 700 nm was measured relative to a blank sample obtained by mixing 5 mL distilled water with 0.5 mL FeCl_3_ 0.1%.

The antioxidant capacity was calculated using the IC50 (half of the antioxidant effect—IC—effective concentration) value (mg/mL), which represents the solution concentration for which the absorbance has a value of 0.5.

Different extract volumes were tested in order to reach the absorbance value of 0.5, due to the variability of plant characteristics and the nonuniformity of phytochemical profiles of plant extracts (experimental values closer to the target value result in more accurate approximation—IC50 for y = 0.5). The optimized values have been set as mentioned above in order to conduct an appropriate comparative study within the same technique and between other methods of assessing the antioxidant activity.

#### 4.5.2. In Silico Methods

##### Molecular Docking

Molecular docking simulations were performed for the identified phytochemicals to assess the potential biological activities on sirtuin isoforms. Crystal structures of human sirtuin 1 (PDB ID: 5BTR, 3.20 Å resolution [68]) and sirtuin 5 (PDB ID: 4HDA, 2.60 Å resolution [69]) in complex with peptide substrates and activator resveratrol, and sirtuin 6 in complex with ADP-ribose and activator quercetin (PDB ID: 6QCD, 1.84 Å resolution), and inhibitor catechin gallate (PDB ID: 6QCJ, 2.01 Å resolution) [21], respectively, were retrieved from RCSB PDB database. Since there are experimentally determined structures available for both SIRT6 activator and inhibitor polyphenolic derivatives, we chose to perform docking experiments on both receptor structures, to discriminate between potential activators and inhibitors.

The preparation of protein structures was performed with YASARA Structure [70], and consisted in the removal of solvent molecule, excepting the structurally relevant water molecules, correction of structural errors and protonation according to the physiological pH (7.4). The structures were further optimized by minimization with AMBER ff14SB forcefield. The validation of the docking protocol was performed by docking the co-crystallized ligands into the active and superposing the predicted pose with the experimentally determined conformation. The ligands used for validation also served as positive controls for docking score comparisons [71,72].

Three-dimensional structures of the tested ligands were generated with DataWarrior 5.2.1 [73]. Ligands were minimized using MMFF94s+ forcefield and protonated at physiological pH. The molecular docking algorithm used was AutoDock Vina v1.1.2, executed within YASARA. The search space (22.5 Å × 22.5 Å × 22.5 Å) was centered around the co-crystallized ligands within the binding sites and 12 docking runs were performed for each ligand.

Docking results were retrieved as the binding energy (ΔG, kcal/mol), predicted dissociation constant (Kd, µM) and ligand efficiency (LE, ΔG\no. of heavy atoms) of the best binding pose for each screened ligand. The conformations of the predicted protein–ligand complexes and molecular interactions were analyzed using BIOVIA Discovery Studio Visualizer (BIOVIA, Discovery Studio Visualizer, Version 17.2.0, Dassault Systèmes, 2016, San Diego, CA, USA).

##### Binding Free Energy Calculations

Short molecular dynamics simulations (1 ns) were performed to estimate the free energy of binding for the positive controls and one promising phytochemical, using the Poisson–Boltzmann (MM/PBSA) method, excluding the entropic term. The simulations of the selected protein–ligand complexes were carried out with YASARA Structure. The simulation system was neutralized by adding Na and Cl ions at 0.9% concentration. Clashes were removed by performing steepest descent and simulated annealing minimizations. AMBER14 force field was used for the protein [74], GAFF2 [75] and AM1BCC [76] for ligand and TIP3P for water. The cut-off for van der Waals forces was 8 Å [77], while the electrostatic forces were treated using the Particle Mesh Ewald algorithm and no cutoff was applied [78]. The integration of motions equations was performed with a multiple timestep of 2.5 fs for bonded and 5 fs for non-bonded interactions at 298 K and 1 atm (isothermal-isobaric ensemble) [79].

### 4.6. Statistical Analysis

Statistical analysis was implemented using the open source software R (R version 4.1.3., R Foundation for Statistical Computing, Vienna, Austria) [80]. The statistics were performed on 5 replicates. Therefore, the application of robust inferential methods becomes vital, especially as they have results with increased accuracy in the case of samples with relatively small sizes [81]. As a means to simultaneously evaluate the effect of two factors, Compound (with three levels: FL and TPC) and Sample (with two levels: common crop and control crop in first analysis or with three levels in second analysis: common crop, Bio crop and NPK crop) on a response variable named Concentration, we used two-way robust ANOVA test for every plant product (peppermint and lemon balm) [82]. Statistical significance is set to 0.05 (5%) and for post hoc analysis a Bonferroni adjusted alpha level was used. Pearson statistical analysis was performed using IBM SPSS Statistics software version 28.0 (IBM Corporation, Armonk, NY, USA). The correlation between antioxidant activity and active principles from vegetal extracts was established by calculating the Pearson correlation coefficient. The Pearson correlation results were interpreted both mathematically and statistically. Interpretations were made after the mandatory application criteria were met (normality of data sets, linearity, independence of measurements and continuity of variables). Moreover, we tested the absence of outliers and we transformed by logarithm in base 10 certain experimental data which did not follow a normal distribution, so that they could be subjected to statistical tests. The significance level was set at 0.05.

## 5. Conclusions

Based on the present studies, we can consider that interventions in the cultivation of medicinal plants can sometimes be beneficial in terms of generating a greater quantity of plant product, but also in enriching the polyphenolic content. The phytosociological cultivation of mint and melissa showed positive effects on the biosynthesis of polyphenolic compounds. Fertilization with organic fertilizer, although not generating a larger quantity of plant raw material, lead to clearly higher polyphenolic contents than in the batches treated with chemical fertilizer. Polyphenols identified and quantified by FT–ICR MS and UHPLC–MS supported the antioxidant activity of assessed plant extracts.

Molecular docking studies supported the hypothesis that the obtained extracts have the potential to directly activate SIRT1, 5 and 6 through several polyphenolic compounds, thus complementing the free radical scavenging activity with the potential stimulation of endogenous antioxidant defense mechanisms.

Future phytosociological studies are needed to investigate the interrelationships between other types of medicinal species belonging to different genera and families.

## Figures and Tables

**Figure 1 plants-11-02398-f001:**
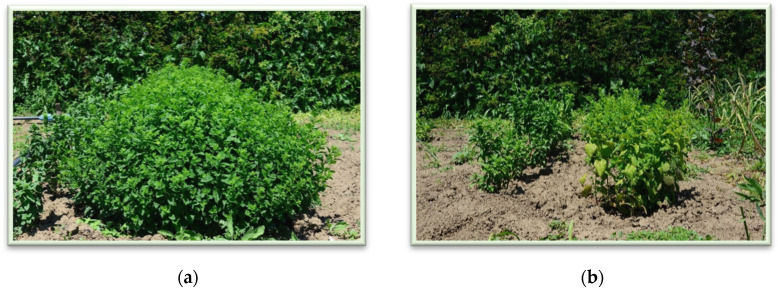
(**a**) MML crops from NPK-fertilized soil; (**b**) MML crops from bio-fertilized soil.

**Figure 2 plants-11-02398-f002:**
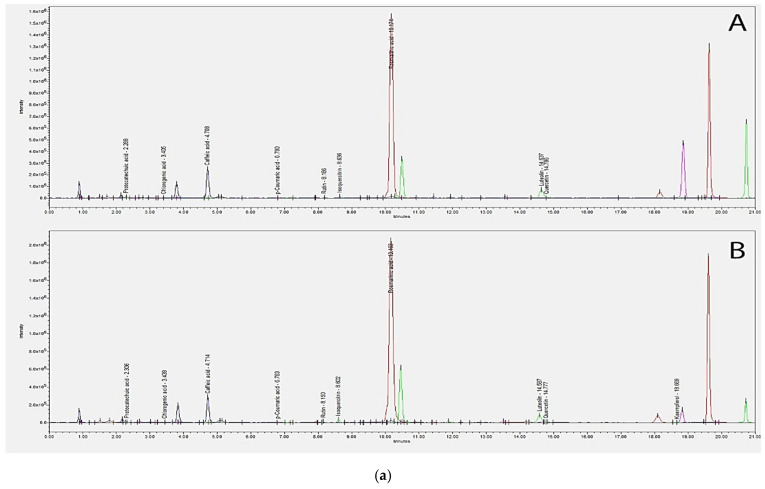
(**a**) *Menthae* extract, entire spectra, ESI−; (A)—MM E and (B)—MF E; (**b**) *Melissae* extract, entire spectra, ESI−; (A)—MLM E and (B)—MLF E. Legend: PRO—Protocatechuic acid, RUT—Rutin, CAF—Caffeic acid, CHL—Chlorogenic acid, LUT—Luteolin, KAE—Kaempferol, ROS—Rosmarinic acid, QUE—Quercetin, ISO—Isoquercitrin, FER—Ferulic acid, COU—*p*-Coumaric acid.

**Figure 3 plants-11-02398-f003:**
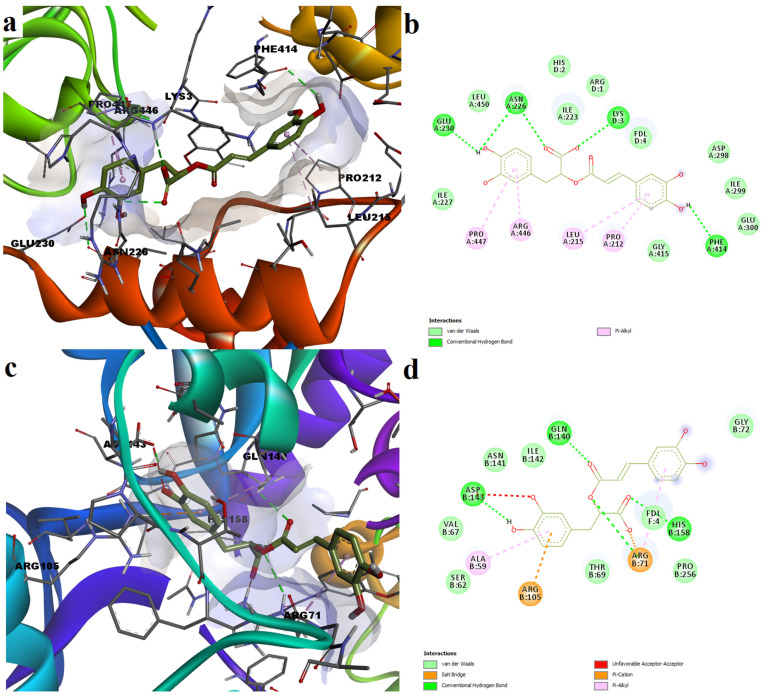
Predicted ligand poses and molecular interactions between rosmarinic acid and SIRT1 and SIRT5. (**a**)—3D conformation of predicted rosmarinic acid-SIRT1 complex; (**b**)—2D representation of protein–ligand interactions for predicted rosmarinic acid–SIRT1; (**c**)—3D conformation of predicted rosmarinic acid–SIRT5 complex; (**d**)—2D representation of protein–ligand interactions for predicted rosmarinic acid–SIRT5 complex.

**Figure 4 plants-11-02398-f004:**
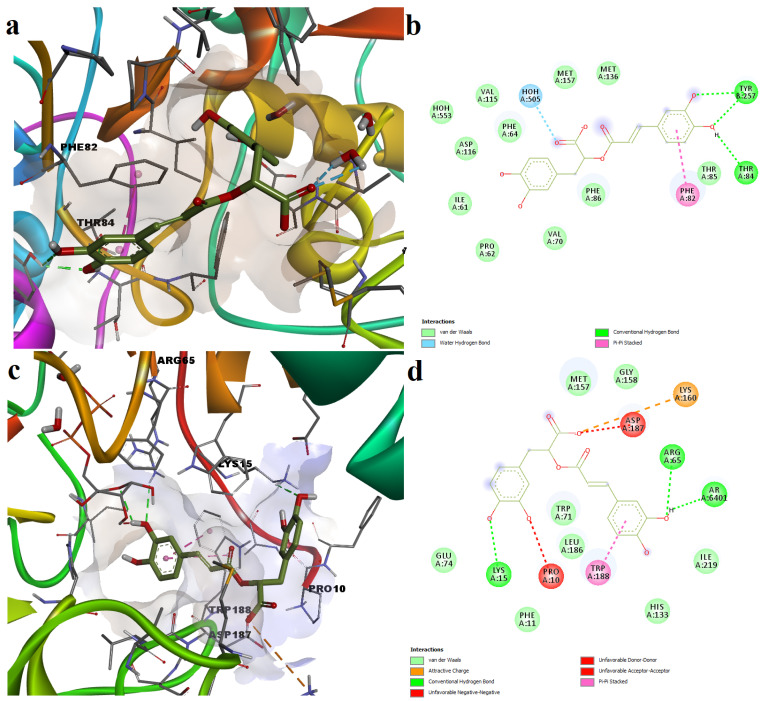
Predicted ligand poses and molecular interactions between rosmarinic acid and SIRT6. (**a**)—3D conformation of predicted rosmarinic acid–SIRT6 complex (activator binding pocket); (**b**)—2D representation of protein–ligand interactions for predicted rosmarinic acid–SIRT6 (activator binding pocket); (**c**)—3D conformation of predicted rosmarinic acid–SIRT6 complex (inhibitor binding pocket); (**d**)—2D representation of protein–ligand interactions for predicted rosmarinic acid–SIRT6 complex (inhibitor binding pocket).

**Table 1 plants-11-02398-t001:** Quantitative analysis of polyphenols from plant material.

Plant Sample	Solvent	TFL (mg/g Eq Expressed in Rutin)	TPC (mg/g Eq Expressed in Tannic Acid)
MM	50% Alcohol	10.11 ± 2.526	91.80 ± 14.828
MF	50% Alcohol	25.87 ± 5.766	101.43 ± 19.329
MLM	70% Alcohol	16.12 ± 2.692	40.90 ± 10.775
MLF	70% Alcohol	22.71 ± 5.160	65.84 ± 28.841

Total flavonoids content (TFL), total phenolic content (TPC). MM—peppermint, control crop; MF—peppermint phytosociological (common) crop; MLM—lemon balm control crop; MLF—lemon balm control (common) crop. Results were expressed as Mean ± SD (*n* = 5).

**Table 2 plants-11-02398-t002:** Quantitative analysis of soils and plant materials.

Identification Probe	ID 103-21	ID 1054	ID 1055	M F	ML F	M NPK	ML NPK	M Bio	ML Bio
pH	8.06	7.38	8.05	-	-	-	-	-	-
HUM [mg/kg]	28.10	36.70	36.10	-	-	-	-	-	-
Res. Cond. [mg/kg]	400.00	1780.00	880.00	-	-	-	-	-	-
N [mg/kg]	1.98	2.73	2.41	37.00	17.60	45.90	26.30	37.60	25.00
P [mg/kg]	403.00	1118.00	660.00	3.60	4.50	4.80	3.60	4.30	4.20
K [mg/kg]	359.00	752.00	464.00	17.80	25.90	21.20	24.00	18.70	24.10
Ca [mg/kg]	-	-	-	21.90	12.10	21.40	12.50	18.30	12.20
Zn [mg/kg]	3.60	7.30	5.90	222.00	360.00	264.00	212.00	219.00	314.00
Cu [mg/kg]	5.90	6.40	5.50	111.00	134.00	111.00	121.00	108.00	130.00
Fe [mg/kg]	12.00	11.20	9.00	1760.00	4720.00	3320.00	2690.00	3840.00	4540.00
Mn [mg/kg]	10.05	15.90	13.90	350.00	254.00	402.00	196.00	380.00	259.00

ID 103-21—control soil sample, ID 1054—NPK fertilized soil sample, ID 1055—BIO fertilized soil sample; M F—mint sample, control crop, unfertilized soil; ML F—lemon balm sample, control crop, unfertilized soil; M NPK—mint sample, NPK fertilized soil; ML NPK—lemon balm sample, NPK fertilized soil; M Bio—mint sample, BIO fertilized soil; ML Bio—lemon balm sample, BIO fertilized soil, HUM—humus, Res. Cond.—determination of electrical conductivity and estimation of total soluble salt content, N—nitrogen, P—phosphorus, K—potassium, Ca—calcium, Zn—zinc, Fe—iron, Mn—manganese.

**Table 3 plants-11-02398-t003:** Quantitative analysis of plant products harvested from fertilized crops.

Plant Extract	Solvent	TFL (mg/g Eq Expressed in Rutin)	TPC (mg/g Eq Expressed in Tannic Acid)
M F	50% Alcohol	25.87 ± 5.766	101.43 ± 19.329
M NPK	50% Alcohol	22.71 ± 5.476	104.17 ± 21.563
M Bio	50% Alcohol	35.38 ± 6.649	120.09 ± 38.467
ML F	70% Alcohol	22.71 ± 5.160	65.84 ± 28.841
ML NPK	70% Alcohol	23.51 ± 6.588	70.26 ± 27.772
ML Bio	70% Alcohol	40.03 ± 5.417	83.41 ± 24.644

M F and ML F—mint and lemon balm from unfertilized soil; M NPK and ML NPK—mint and lemon balm from fertilized common crop obtained with chemical fertilizer; M Bio and ML Bio—mint and lemon balm from fertilized common crop obtained with biological fertilizer.

**Table 4 plants-11-02398-t004:** Quantitative analysis of polyphenols in plant extracts.

Plant Extract	TFL (mg/g Eq Expressed in Rutin)	TPC (mg/g Eq Expressed in Tannic Acid)
MM E	54.70 ± 10.995	327.46 ± 3.003
MF E	86.78 ± 10.996	411.73 ± 13.696
MLM E	65.38 ± 15.772	333.67 ± 34.451
MLF E	78.74 ± 8.055	574.54 ± 45.203

Total flavonoids content (TFL), total phenolic content (TPC). Results were expressed as Mean ± SD (*n* = 5). MM E—peppermint extract, control crop; MF E—peppermint extract, phytosociological (common) crop; MLM E—lemon balm extract, control crop. MLF E—lemon balm extract, phytosociological (common) crop.

**Table 5 plants-11-02398-t005:** Polyphenols found in sample extracts identified by FT–ICR MS.

Sample Name	*m*/*z*		MM E		MF E		MLM E		MLF E	
	ESI+	ESI−	ESI+	ESI−	ESI+	ESI−	ESI+	ESI−	ESI+	ESI−
PRO	-	153.02	-	+	-	+	-	+	-	+
RUT	611.16	609.15	-	+	-	+	+	+	+	+
CAF	181.05	179.03	+	+	+	+	+	+	+	+
CHL	355.10	353.09	+	+	+	+	+	+	+	+
LUT	287.06	285.04	+	+	+	+	+	+	+	+
KAE	287.06	285.04	+	+	+	+	+	+	+	+
ROS	361.09	359.08	+	+	+	+	+	+	+	+
QUE	303.05	301.04	+	+	+	+	+	+	+	+
ISO	465.10	463.09	+	+	+	+	+	+	+	+
FER	195.07	193.05	+	+	+	+	+	+	+	+
COU	165.05	163.04	-	+	-	+	+	+	+	+

PRO—Protocatechuic acid, RUT—Rutin, CAF—Caffeic acid, CHL—Chlorogenic acid, LUT—Luteolin, KAE—Kaempferol, ROS—Rosmarinic acid, QUE—Quercetin, ISO—Isoquercitrin, FER—Ferulic acid, COU—*p*-Coumaric acid, *m*/*z*—atomic mass; MM E—peppermint extract, control crop; MF E—peppermint extract, phytosociological (common) crop; MLM E—lemon balm extract, control crop; MLF E—lemon balm extract, phytosociological (common) crop.

**Table 6 plants-11-02398-t006:** Polyphenol concentrations found in extracts (μg compound/g extract) identified by UHPLC-MS quantification.

Sample Name	MM E	MF E	MLM E	MLF E
PRO [μg/g]	57.13 ± 1.883	48.62 ± 2.275	78.82 ± 1.910	98.54 ± 1.985
RUT [μg/g]	27.36 ± 2.117	68.00 ± 2.058	28.74 ± 2.256	205.82 ± 1.309
CAF [μg/g]	321.44 ± 1.727	345.45 ± 2.221	84.15 ± 2.146	1296.55 ± 1.911
CHL [μg/g]	56.09 ± 1.911	75.42 ± 2.385	73.99 ± 1.995	397.91 ± 2.237
LUT [μg/g]	99.68 ± 2.225	87.03 ± 1.652	121.19 ± 1.723	547.50 ± 1.866
KAE [μg/g]	M	0.94 ± 0.148	1.22 ± 0.180	3.13 ± 0.689
ROS [μg/g]	43.95 ± 2.145	51.02 ± 2.080	48.59 ± 2.143	64.31 ± 1.750
QUE [μg/g]	5.93 ± 1.205	116.38 ± 2.100	4.62 ± 0.7605	76.84 ± 1.722
ISO [μg/g]	176.48 ± 2.355	170.67 ± 2.165	170.67 ± 4.455	270.91 ± 2.108
FER [μg/g]	M	M	M	M
COU [μg/g]	28.81 ± 1.888	20.78 ± 1.722	10.32 ± 1.624	54.64 ± 1.641

PRO—Protocatechuic acid, RUT—Rutin, CAF—Caffeic acid, CHL—Chlorogenic acid, LUT—Luteolin, KAE—Kaempferol, ROS—Rosmarinic acid, QUE—Quercetin, ISO—Isoquercitrin, FER—Ferulic acid, COU—*p*-Coumaric acid, M—missing; MM E—peppermint extract, control crop; MF E—peppermint extract, phytosociological (common) crop; MLM E—lemon balm extract, control crop; MLF E—lemon balm extract, phytosociological (common) crop, results were expressed as Mean ± SD (*n* = 3).

**Table 7 plants-11-02398-t007:** Determination of antioxidant activity.

Sample	DPPH IC50 (mg/mL)	95%CI	ABTS IC50 (mg/mL)	95%CI	FRAP IC50 (mg/mL)	95%CI
MM E	0.082	0.079–0.086	0.037	0.035–0.038	0.807	0.754–0.870
MF E	0.056	0.052–0.059	0.028	0.026–0.030	0.619	0.585–0.658
MLM E	0.048	0.040–0.055	0.027	0.024–0.029	0.756	0.689–0.836
MLF E	0.042	0.035–0.047	0.025	0.024–0.026	0.591	0.528–0.672
Vitamin C	0.016	0.0160–0.0169	–*		–*	
Trolox	–*		0.033	0.028–0.037	–*	
FeSO_4_	–*		–*		0.102	0.098–0.106

DPPH: 2,2-diphenyl-1-picryl-hydrazine; ABTS: 2,20-azinobis-3-ethylbenzotiazoline-6-sulfonic acid; FRAP: ferric reducing antioxidant power, 95%CI—95% confidence interval of IC50, “–*” = the standard has not been used for this method.

**Table 8 plants-11-02398-t008:** Molecular docking results for selected sirtuin isoforms.

	SIRT1 (Activator)		SIRT5 (Activator)		SIRT6 (Activator)		SIRT6 (Inhibitor)	
Ligand	ΔG (kcal/mol)	LE	ΔG (kcal/mol)	LE	ΔG (kcal/mol)	LE	ΔG (kcal/mol)	LE
Caffeic acid	−7.186	0.553	−6.224	0.479	−6.274	0.483	−6.411	0.493
Chlorogenic acid	−8.027	0.321	−7.212	0.289	−6.609	0.264	−7.262	0.291
Ferulic acid	−7.334	0.524	−6.057	0.433	−5.721	0.409	−6.172	0.441
Isoquercitrin	−8.995	0.273	−8.205	0.249	−6.460	0.196	−5.865	0.178
Kaempferol	−9.827	0.468	−7.583	0.361	−6.690	0.319	−6.787	0.323
Luteolin	−9.520	0.453	−7.962	0.379	−6.419	0.306	−7.279	0.347
*p*-Coumaric acid	−7.376	0.615	−5.720	0.477	−6.977	0.581	−5.900	0.492
Protocatechuic acid	−5.858	0.533	−5.750	0.523	−6.232	0.567	−5.703	0.519
Quercetin	−9.259	0.421	−7.554	0.343	−6.899 *	0.314 *	−6.912	0.314
Rosmarinic acid	−9.638	0.371	−8.357	0.321	−7.775	0.299	−8.287	0.319
Rutin	−9.709	0.226	−8.764	0.204	−7.623	0.177	−7.940	0.185
Resveratrol *	−9.332	0.549	−5.220	0.307	-	-	-	-
Catechin gallate *	-	-	-	-	-	-	−8.732	0.273

ΔG—binding energy; LE—ligand efficiency; *—positive control.

## Data Availability

Not applicable.

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
