# Peer review of "The Influence of Phytosociological Cultivation and Fertilization on Polyphenolic Content of Menthae and Melissae folium and Evaluation of Antioxidant Properties through In Vitro and In Silico Methods"

_plants, 2022, doi:10.3390/plants11182398_

Round 1
Reviewer 1 Report (Previous Reviewer 3)
Manuscript in corrected form can be accepted for publication.
Author Response
Thank you for your appreciation. Moreover, English was also extensively edited.
Reviewer 2 Report (Previous Reviewer 4)
The authors addressed most of the required modifications. However, there are still a couple of flaws:
- In Table 3 the samples are still indicated as "Vegetal plant", pease change in "Plant extract";
- Table 6 and 7 are missing the standard deviations and the number of replicates.
Once these modifications have been made, the manuscript is suitable for publication on "Plant".
Author Response
The authors addressed most of the required modifications. However, there are still a couple of flaws:
- In Table 3 the samples are still indicated as "Vegetal plant", pease change in "Plant extract";
We apologize for the error; corrections were made as suggested ( line185).
- Table 6 and 7 are missing the standard deviations and the number of replicates.
Missing data was added, as suggested (lines 245,287)
Once these modifications have been made, the manuscript is suitable for publication on "Plant".
Thank you for your appreciation. Moreover, English was also extensively edited.
This manuscript is a resubmission of an earlier submission. The following is a list of the peer review reports and author responses from that submission.
Round 1
Reviewer 1 Report
The objectives of the study were to monitor the mint and melissa crops from common batches and to compare with single-component batches in terms of variation in polyphenol content. The polyphenolic derivatives were quantified by spectrophotometric and HPLC methods, and their antioxidant bioactivities were determined in vitro. In addition, this study also investigated the potential stimulatory activity of the detected polyphenols on three sirtuin isoforms (SIRT1, SIRT5 and SIRT6) by using molecular docking simulations. Although I appreciated authors' efforts, this manuscript displayed various technical deficiencies. The present results were preliminary and would not show any significant impacts to the natural medicines related research fields. In brief, this manuscript is not recommended to accept for publication in Plants. In addition, there are several major comments to be addressed as following.
1. There were various serious typographic, grammar, and format errors observed in this manuscript. Some words in title should be in italics. Page 3, lines 116-117, some sentences may be missed. Authors have to check and revise these errors very carefully.
2. Authors seemed to be confused with FT-ICR and FT-IR. Careful revisions should be performed throughout the manuscript. Table 9, the “retention time” was wrong.
3. The experimental results provided by colorimetric method were not so interesting since there were a lot of interferences influencing the obtained data. It meant that the determination of total flavonoids, phenolic acids, and phenolics may not be totally contributed by the indicated compounds.
4. The antioxidant activities were only examined in the extracts level and no any molecular data were incorporated. In addition, there were not any positive control data included in Table 11. Compared with the standard used (page 27, line 749), the DPPH activities of extracts were not significant.
5. In the molecular docking section, authors should provide some more experimental data to evidence their prediction results if they wish to resubmit this manuscript.
6. In the References section, the writing manner of several references did not follow the style of this journal. Authors have to check and revise these errors carefully.
Reviewer 2 Report
Comments attached.

Reviewer 3 Report
In a manuscript entitled: "The Determination of Polyphenolic Content in Menthae and Melissae folium Samples from Common Cultures (Phytosocio-logical)", the group of authors tried to assess the effect of combining the cultivation of two plants on the level of secondary metabolites. The idea is interesting but mixed crops will certainly make it difficult to harvest plants separately.
The paper presents a very wide scope of research, which differed significantly from the purpose of the research. For this reason, the work is chaotic and cannot be published in its present form.
Here are the arguments that they disqualify this manuscript for publication
- The title of the work does not reflect its content
- In addition to the main thread of the manuscript, there are additional contents that could be the theme of a separate work: The effect of fertilization on the level of secondary metabolites in the leaves of mint and lemon balm. In turn, the part on molecular docking studies is the most interesting and valuable part of the manuscript, it should be emphasized both in the title of the work and in the abstract.
- The manuscript is too extensive and many figures and tables are redundant here and could be successfully included in the Supplementary material or as material available from the authors at the reviewer's request (mainly all spectra from the MS analysis and Table 13 and 14 as well as Fig. 16)
- Many results are presented without statistical data analisys (Tables: 1, 5, 6, 11)
- Some tables can be combined (Table 2 with 3 and Tables 7, 8 and 9)
- Part of disscussion is linked to the results (for the results of anti-radical activity and molecular docking. But separate Disscusion section is also added.
- There are different units of TFL, TPCA and TPC ones it is g / 100g, another time mg / g.
Due to so many shortcomings, I believe that the work in its current form is not suitable for publication
Reviewer 4 Report
The manuscript “The Determination of Polyphenolic Content in Menthae and Melissae folium Samples from Common Cultures (Phytosociological)” deals with the comparison between single and common (phytosociological) batches of mint and melissa crops in terms of the variation in polyphenol content (flavones, PCA’s -phenolcarboxylic acids and total polyphenols). The authors also studied the comparison between polyphenol content in plant products from batches supplemented with biofertilizer and chemical fertilizers to determine their antioxidant action in vitro. The potential stimulatory activity of the detected polyphenols on three sirtuin isoforms (SIRT1, SIRT5 and SIRT6) by using molecular docking simulations was also investigated.
The subject addressed by this study is of great interest and falls within the scope of the journal. However, there are some critical issues to be addressed:
-In “Results”, the authors comment on the higher antioxidant capacity of the extracts obtained from the phytosociological crops compared to that of the “control” plants. However, the example they give is the content of protocatecuic acid, which, in Table 10, is reported to be lower in the phytosociological mint crop extracts than in the “control” ones. Can you explain this result? Also, both Table 10 and Table 11 are missing the standard deviations and the numbers of replicates.
- In “Discussion” (lines 456-458), the authors write that the extraction of polyphenols from plant products was made by using ethanol 50 [%] for mint and ethanol 70 [%] for lemon balm, corresponding to what is reported in Table 1. On the other hand, in Table 5 the use of ethanol 50% or ethanol 70% depends on the type of fertilization (chemical or biological) and not on the type of plant. Is this a typo?
The following minor flaws should also be addressed:
-Abstract line 22: Menthae piperita folium L. and Melissae officinalis folium L. (it is the first time you mention the name of the species)
-Line 24: write out full name for PCA the first time you mention
-Line 28: “…led to the biosynthesis of a larger amount of polyphenols compared to that of…”
-Lines 116, 117: ?
-Table 1: vegetal plant? (plant is ok) Also in Table 5
-Line 158: fertilized or fertilised
-Line 562: “ and AlCl3. From our obtained extractive…”.
-Line 562-564: Please clarify